# Clinician Perspectives of Communication with Aboriginal and Torres Strait Islanders Managing Pain: Needs and Preferences

**DOI:** 10.3390/ijerph19031572

**Published:** 2022-01-29

**Authors:** Christina M. Bernardes, Stuart Ekberg, Stephen Birch, Renata F. I. Meuter, Andrew Claus, Matthew Bryant, Jermaine Isua, Paul Gray, Joseph P. Kluver, Daniel Williamson, Corey Jones, Kushla Houkamau, Marayah Taylor, Eva Malacova, Ivan Lin, Gregory Pratt

**Affiliations:** 1Aboriginal and Torres Strait Islander Health Research Program, QIMR Berghofer Medical Research Institute, Brisbane, QLD 4006, Australia; corey.jones@qimrberghofer.edu.au (C.J.); Kushla.houkamau@qimrberghofer.edu.au (K.H.); gregory.pratt@qimrberghofer.edu.au (G.P.); 2School of Psychology and Counselling, Queensland University of Technology, Brisbane, QLD 4059, Australia; stuart.ekberg@qut.edu.au (S.E.); r.meuter@qut.edu.au (R.F.I.M.); 3Centre for the Business and Economics of Health, The University of Queensland, Brisbane, QLD 4072, Australia; stephen.birch@uq.edu.au; 4Tess Cramond Pain and Research Centre, Metro North Hospital and Health Service, Brisbane, QLD 4021, Australia; andrew.claus@health.qld.gov.au (A.C.); paul.gray@health.qld.gov.au (P.G.); 5North Queensland Persistent Pain Management Service, Townsville Hospital and Health Service, Townsville, QLD 4814, Australia; matthew.bryant2@health.qld.gov.au (M.B.); marayah.taylor@health.qld.gov.au (M.T.); 6Cultural Capability Services, Queensland Health, Brisbane, QLD 4000, Australia; jermaine.isua@health.qld.gov.au; 7Aboriginal and Torres Strait Islander Health Division, Queensland Health, Brisbane, QLD 4000, Australia; daniel.williamson@health.qld.gov.au; 8Persistent Pain Clinic, Metro South Hospital and Health Service, Brisbane, QLD 4102, Australia; joseph.kluver@health.qld.gov.au; 9Statistics Group, QIMR Berghofer Medical Research Institute, Brisbane, QLD 4006, Australia; eva.malacova@qimrberghofer.edu.au; 10Western Australian Centre for Rural Health (WACRH), The University of Western Australia, Geraldton, WA 6531, Australia; ivan.lin@uwa.edu.au

**Keywords:** communication, training, Aboriginal, Torres Strait Islander, clinician, needs

## Abstract

Poor communication is an important factor contributing to health disparity. This study sought to investigate clinicians’ perspectives about communicating with Aboriginal and Torres Strait Islander patients with pain. This multi-site and mixed-methods study involved clinicians from three pain management services in Queensland, Australia. Clinicians completed a survey and participated in focus groups. Clinicians rated the importance of communication training, their knowledge, ability, and confidence in communicating with Aboriginal and Torres Strait Islander patients using a 5-point Likert scale. Rating scores were combined into low (scores 1–2); moderate (score 3) and high (scores 4–5). Informed by an interpretive description methodology, thematic analysis of focus group data was used to identify the communication needs and training preferences of clinicians. Overall (*N* = 64), 88% of clinicians rated the importance of communication training when supporting Aboriginal and Torres Strait Islander patients as “high”. In contrast, far fewer clinicians rated as “high” their knowledge (28%), ability (25%) and confidence (28%) in effectively communicating with Aboriginal and Torres Strait Islander patients. Thematic analysis identified three areas of need: knowledge of Aboriginal and Torres Strait Islander cultures, health beliefs, and understanding cross-cultural cues. Communication skills can be learned and training, in the form of a tailored intervention to support quality engagement with Aboriginal and Torres Strait Islander patients, should combine cultural and communication aspects with biomedical knowledge.

## 1. Introduction

Persistent pain (i.e., pain lasting >3 months) commonly involves a combination of physical, psychological and social distress, which makes it challenging to manage [1,2]. People with persistent pain have difficulty sleeping, going to work or social activities and report higher rates (i.e., five times as likely to report their daily activities were “limited a lot”) of limitations to daily activities than those without pain [3]. Additionally, persistent pain costs the Australian economy approximately 73 billion AUD per annum, comprised of 48.3 billion AUD in lost productivity, 12.2 billion AUD in direct health system costs, and indirect costs such as informal care, welfare payments and aids (among others). Persistent pain impacts more than 68% of Australians of working age and accounts for 6.8% of the total burden of disease in Australia and 6.5% of total health system expenditure [3].

In a study involving an Aboriginal community, Honey and Jacobs (1996) found that nearly half the adults in the community experienced lower back pain; although it impacted people’s lives, they did not express public pain or illness behaviours as recognised by the European Australian experience. Moreover, many participants would describe their pain to a non-clinical researcher, but would not necessarily do so when interacting with their doctor or other Aboriginal people [4]. In Australia, Canada and the USA, the burden of persistent pain amongst Indigenous people compared to non-Indigenous populations is now known to be disproportionally high [5,6,7] and such that reducing the burden of persistent pain amongst Indigenous people is internationally a priority [5].

Access to pain management is a fundamental human right and a health priority [8,9,10], however, access to and the quality of care may be lower for Aboriginal and Torres Strait Islander patients than for non-Indigenous patients [6,11]. In Australia, the number of Aboriginal and Torres Strait Islander patients who access pain management services is not commensurate with either population proportion or need. For example, although Aboriginal and Torres Strait Islander people constitute 9% of the overall population of North Queensland, only 3% of patients who accessed the North Queensland Persistent Pain Management Service identify as Aboriginal and Torres Strait Islanders [11]. To date, little is known about the way persistent pain management services can address this disparity and improve care for Aboriginal and Torres Strait Islander patients.

There are many barriers to Aboriginal and Torres Strait Islander patients being able to access health care for pain-related conditions including historical, societal, health system, health service, and clinician level factors [12]. These factors may include Aboriginal and Torres Strait Islander peoples’ distrust of health care services, due to poor historical and ongoing treatment [2,13]; lack of culturally appropriate tools to measure pain [2]; and inadequate communication and cultural awareness skills among clinicians, which restrict clinicians’ ability to recognise and interpret culturally specific ways of expressing and managing pain [13,14,15]. In addition, there is increasing evidence that poor communication between patients and clinicians is an important factor influencing access to health services [16]. Meeting these communication challenges requires improving the cultural capability of clinicians; leading to improved access to quality care for Aboriginal and Torres Strait Islander patients. This study focused on cultural capability as defined in the Aboriginal and Torres Strait Islander Cultural Capability framework “cultural capability refers to skills, knowledge and behaviours that are required to plan, support, improve and deliver services in a culturally respectful and appropriate manner” [17].

Pain is a complex experience, with characteristics that are particularly difficult to translate into language [18]. Even patients with effective language and social skills can be challenged to describe the complexities of the multidimensional experiences in a direct and representative manner [18]. Cultural and social factors, including beliefs, customs, languages, and relationships with self and society are known to influence how a person experiences pain [2] and often challenge clinicians’ ability to sensitively interpret, assess, and manage pain [16,19]. Ineffective communication with health professionals is a major reason for Aboriginal people with musculoskeletal pain “giving up” on the healthcare system [20,21]. Communication failures included the use of medical jargon that was not understood by patients or communicating in ways that were not congruent with Aboriginal peoples’ experiences [7,16]. Aboriginal patients have reported their struggle to describe their pain to clinicians, finding it difficult to make themselves understood and also to understand the recommendations given to them [21]. Some patients have reported negative attitudes from clinicians, such as stereotyping or not having sufficient time in consultations [21]. In contrast, clinicians were perceived positively if they were known to the community, attentive listeners and showed broader interest in patients’ lives [20]. Notwithstanding these challenges, clinicians’ communication skills are modifiable, so communication training is a vital part of ensuring adequate skills among clinicians [22,23,24].

One way of improving the communication is through a tailored training intervention grounded in Aboriginal and Torres Strait Islander experience and knowledge that addresses knowledge gaps reported by clinicians [25]. Accordingly, to guide the development of a tailored training intervention to support better communication and care for Aboriginal and Torres Strait Islander patients managing pain, this study aimed to (1) assess clinicians’ perceptions of their knowledge, ability and confidence in communicating with Aboriginal and Torres Strait Islander patients; and (2) identify clinicians’ communication needs and preferred modes of training.

## 2. Materials and Methods

### 2.1. Study Design

The present study is the first stage of a multi-centre interventional study to improve communication between clinicians and Aboriginal and Torres Strait Islander patients accessing persistent pain management services in Australia. The training intervention (to be rolled out as part of a larger project), is a tailored training package for clinicians that includes both cultural (“cultural capability”) and communication (“clinical yarning”) learning modules. Cultural capability refers to the skills, knowledge and behaviours that are required to plan, support and deliver services in a respectful and appropriate manner [17], and is a key component of culturally safe healthcare [26,27]. Clinical yarning is a patient-centred approach that combines Aboriginal cultural communication preferences with biomedical understandings of health and disease [25].

This study used a mixed-methods study design. For the quantitative data, a custom-designed survey (Appendix A) was used to assess the extent to which clinicians perceived communication training to be important, their knowledge of, ability and confidence to communicate with Aboriginal and Torres Strait Islander patients. Qualitative data were gathered through focus groups/research yarns with clinicians to explore communication needs and training preferences. Analysis of data from Aboriginal and Torres Strait Islander consumers is the focus of a separate manuscript currently under development.

### 2.2. Study Setting and Participants

In Queensland, there are six (6) persistent pain clinics within publicly-funded hospitals and health services, including one pediatric service. These clinics are hospital-based and provide a combination of in-patient and out-patient pain management services. Aboriginal and Torres Strait Islander patients represent between 4% and 8% of the total number of patients supported by the three services involved in this study.

Participants were clinicians (medical doctors, nursing staff and allied health) working in three public persistent pain services (two metropolitan and one regional service) in Queensland, in October 2020. All clinicians were invited to complete a cultural awareness and communication needs survey and participate in focus groups.

Flyers were used to advertise the study. Written consent was obtained from all participants.

### 2.3. Data Collection

Cultural awareness and communication needs survey (CACNS)—Clinicians were invited to complete a cultural awareness and communication needs survey. This customised survey was based on a survey previously designed at the University of Western Australia [28], as part of a project that developed the Clinical Yarning framework that is also used in the present study. This survey is not a validated tool and includes standard items identified in the literature as essential for effective communication, i.e., attitudes (behaviour indicators) and values connected to the importance attributed by clinicians to communication training, knowledge, ability and confidence (Appendix A). Previous research suggested that the effectiveness of training programs is correlated with the importance placed on communication training by clinicians, the perceived benefits of learning opportunities for professional development, and the extent to which the training package utilises a variety of educational methods [29,30,31]. The CACNS survey included questions relating to demographic characteristics and location (i.e., study site, profession, age, sex, and Indigenous status)*,* participation in previous cultural training (Yes/No)*,* types of training received, and a communication skills and cultural awareness section in which clinicians rated the *importance of communication training* when working with Aboriginal and Torres Strait Islander patients and self-assessed their *knowledge, ability* and *confidence* when communicating with Aboriginal and Torres Strait Islander patients with persistent pain (1= *very low to 5= very high*).

*Focus groups*—In addition to quantitative data generated through CACNS, qualitative data were generated through focus groups. A purposive sample of clinicians of mixed professions was involved in focus groups at each study site. Clinicians working together at each study site were asked to participate in the focus groups by the local investigator via either an email or face-to-face invitation. The focus groups were facilitated by an Aboriginal researcher (GP) and a Torres Strait Islander cultural capability facilitator (JI), supported by the research team (which included two Aboriginal research assistants (KH and MT); and an Aboriginal communication researcher (SE)), using a semi-structured interview guide (Appendix A). The interview guide was structured to explore barriers and enablers for three key elements of clinical communication (1) building rapport, (2) gathering background information, and (3) developing a treatment/management care plan. Participants were also asked to identify their communication needs and training preferences. Qualitative data from the focus groups were audio-recorded and notes were taken by two research team members (CMB and CJ) during the discussions. These notes and transcripts were imported to Microsoft Excel for analysis.

### 2.4. Data Analysis

#### 2.4.1. Quantitative Data—Cultural Awareness and Communication Needs Survey

All data were analysed using IBM SPSS Statistics for Windows, version 23 (IBM Corp., Armonk, NY, USA). Categorical demographic variables were summarised using counts and percentages. Normally distributed continuous variables were summarised using means and standard deviation. Statistical significance was set at alpha = 0.05. Categorical variables were compared using a Chi-squared test or Fisher’s exact test, and all *p*-values were two-sided.

For each item (4) of the communication skills and cultural awareness section, scores were combined into *low* (scores 1–2: *very low or low); moderate* (score 3: *moderate*) and *high* (scores 4–5: *high or very high*). Reported are the overall proportions of clinicians, medical doctors and nursing staff (two main subgroups in the sample) who rated “*high*” (i.e., “*scores 4–5*”) the importance of communication training, and their knowledge, ability and confidence to effectively communicate with Aboriginal and Torres Strait Islander patients. The study sites were grouped according to location as metropolitan (study sites A and C) or regional (study site B).

#### 2.4.2. Qualitative Data—Focus Groups

This research adopted an interpretive description methodology, which is aligned with a naturalistic orientation to inquiry [32]. Interpretive description investigates a clinical phenomenon of interest to the discipline for the purpose of identifying thematic patterns within and across subjective experiences and generating interpretive descriptions of these capable of informing clinical practice [32,33]. This approach is congruent with the aims of this study for two key reasons. First, it ensures a focus on individuals’ experiences of communication (e.g., clinicians’ more or less effective forms of communication with Aboriginal and Torres Strait Islander patients). Second, this approach focuses on generating findings that are relevant to practice (e.g., identifying clinicians’ needs and preferences for communication training). Transcripts were reviewed several times to allow researchers to familiarise themselves with the data. The data were initially examined to identify barriers and enablers to communication needs and training preferences [34].

Inductive thematic analysis was undertaken to identify candidate thematic patterns [35,36]. Analysis was performed independently by the first (CMB) and second authors (SE), and consensus reached by discussion between all authors. Themes were cross-checked to ensure full agreement. The data analysis identified a number of relevant subthemes across the three study sites. In this article, we report the needs and training preferences of clinicians.

### 2.5. Researchers’ Characteristics and Reflexivity

The research team is composed of researchers of diverse academic disciplines, including medicine, nursing, psychology and physiotherapy. The team includes both clinicians (9) and non-clinician (7) researchers. The team is also comprised of researchers from diverse cultural and linguistic backgrounds. Five of the authors are of Aboriginal or Torres Strait Islander backgrounds and all the remaining authors are non-Indigenous. The research team composition promoted reflexivity across the research process, by including Aboriginal and Torres Strait Islander as well as non-Indigenous perspectives, clinical and non-clinical perspectives, and multidisciplinary perspectives.

## 3. Results

### 3.1. Cultural Awareness and Communication Needs Survey

#### 3.1.1. Demographics of Participants

Sixty-four (71%) clinicians out of a total of 90 clinicians working at the three study sites completed the cultural awareness and communication needs survey. The majority of clinicians were female (67%; mean age = 43 (SD = 11.64)) and more than a third (36%) were either pain medicine specialists or pain medicine registrars (Table 1). Most of the clinicians (73%) reported participating in previous cultural training, most commonly compulsory work-based training (approximately <1 h presentation or online training), or during university studies.

#### 3.1.2. Communication Skills and Awareness Ratings

Perspectives about the importance of communication training, knowledge of how to effectively communicate, and ability and confidence to communicate varied across the study sites (Table 2) and professions (Figure 1).

Overall, a large proportion of clinicians (88%) rated as “*high*” the importance of communication training when working with Aboriginal and Torres Strait Islander patients. However, there was a much lower proportion of clinicians who reported as “*high*” their knowledge (28%), ability (25%), and confidence (28%) to effectively communicate with Indigenous patients (Table 2).

There was a significant difference between metropolitan and regional sites with respect to the importance of communication, knowledge, ability and confidence to communicate effectively with Aboriginal and Torres Strait Islander patients (*p* < 0.05). A higher proportion of clinicians at the regional site compared to the clinicians at the metropolitan sites rated as “*high*” the importance of communication training (100% vs. 77%; *p* = 0.019); knowledge (35% vs. 23%; *p* = 0.025); ability (38% vs. 14%; *p* = 0.025) and their confidence (41% vs. 17%; *p* = 0.019) to communicate with Aboriginal and Torres Strait Islander patients (Appendix A).

At site A—metropolitan, although half (50%) of the medical doctors rated both the importance and their knowledge of effectively communicating with Aboriginal and Torres Strait Islander patients as *“high*”, only a quarter (25%) rated their ability as “*high*”, and none (0%) rated their confidence to communicate with Aboriginal and Torres Strait Islander patients as “*high*” (Figure 2).

At site B—regional, all medical doctors (100%) rated the importance of communication as “*high*”; however, a quarter (25%) of medical doctors rated their knowledge as “*high*”, and their ability to communicate as “*high*” and only a third (33%) rated their confidence to communicate with Aboriginal and Torres Strait Islander patients as “*high*”.

At site C—metropolitan, although a majority of medical doctors (80%) rated the importance of communication as “*high*”, less than a quarter (20%) rated their knowledge and confidence to communicate with Aboriginal and Torres Strait Islander patients as “*high*”. None (0%) of the medical doctors rated their ability to communicate with Aboriginal and Torres Strait Islander patients as “*high*”.

Across the three study sites, a high proportion of the nursing staff (79%) rated the importance of communication training as “*high*”. However, less than a quarter (18%) of nursing staff rated their knowledge, ability and confidence in effectively communicating with Aboriginal and Torres Strait Islander patients as “*high*” (Figure 3).

### 3.2. Focus Groups

#### 3.2.1. Demographics of Participants

Twenty clinicians (22%, Site A—Metropolitan *N* = *5*; Site B—Regional *N* = 7 and Site C—Metropolitan *N* = 8; respectively) out of 90 agreed to be involved in the focus group discussions. The time clinicians worked in the pain clinic varied highly across study sites; the median time was 3 years (range from 3 months to up to 30 years). The majority of participants were female (65%) and the mean age was 41 years (SD = 10.92). Thirty percent were pain medicine specialists (Table 3).

#### 3.2.2. Findings


*Communication Needs*


Overall, most clinicians reported unmet needs relating to cultural education. Clinicians identified three areas of need: (1) knowledge of Aboriginal and Torres Strait Islander cultures, (2) knowledge of health beliefs and (3) the ability to recognise cultural-specific communication practice.

Clinicians reported feeling anxious and a fear of asking, doing or saying something inappropriate.

C8: “You can get caught up in your head worrying about being offensive or just using the wrong words or making a mistake that is the thing that causes them [Aboriginal and Torres Strait Islander patients] to want to go somewhere else or not seek help at all”.(Clinician at a metropolitan service)

Clinicians also reported uncertainty about asking if somebody identifies as an Aboriginal and Torres Strait Islander Australian and their family origin.

C4: “I don’t know if this is right or wrong but I wish someone had have told me that it’s okay to ask if somebody identifies [as an Aboriginal and Torres Strait Islander] and where there family is or where they’re from and not be afraid to ask”.(Clinician at a metropolitan service)

Clinicians indicated uncertainty about aspects of Aboriginal and Torres Strait Islander cultures known as men’s and women’s business, what they should know and whether and how they should talk to patients about this.

C6: “I guess sometimes I feel like I assume what’s going to be okay to ask a female patient versus a male patient. Maybe if we were confident in that a little bit more I think that would you know I guess feel a little bit more competent and comfortable asking particular questions if we had a bit better understanding about what’s appropriate to ask I suppose”.(Clinician at the regional service)

Clinicians acknowledged their lack of cultural knowledge and a desire to have a better understanding of the local context and the history of Aboriginal and Torres Strait Islander people. For example, in the following quote, a participant adds to a comment that was made by one of the facilitators about culture and illness.

C11: “I think a local context of history is really useful and just understanding and fleshing out your comments that you’ve made about, “It’s not our culture that makes us sick”, what that means and you know the facts of history and politics and why we are where we are. I find those things as you say confronting but really important to put out there in the middle and acknowledged”.(Clinician at the regional service)

It was recognised that experiences across metropolitan, regional, and remote communities are different and that having an Aboriginal and Torres Strait Islander staff member who is aware of local cultures and histories could help to clarify these differences.

C1: “I think forming that relationship with your health workforce; your Indigenous health workforce is really key as well. Because some of those health workers and some of those liaison officers are actually traditional owners and they hold a community authority, which is really valuable in terms of connecting with our consumers and clients”.(Clinician at a metropolitan service)

Cultural beliefs about health, spirit and body held by Aboriginal and Torres Strait Islander patients are poorly understood by clinicians.

C6: “Like one thing that I like I think and this is going back ages ago was that, and I might be wrong, that Aboriginal and Torres Strait Islanders don’t like to come to hospital because there’s that fear that if they don’t go back home, is that right? Then their spirit doesn’t rest? Or is that not right?”.(Clinician at the metropolitan service)

In their practice, clinicians reported having situations where they would have benefited from cultural guidance on how to broach and discuss with patients the impact of psychological trauma on health and the body, treatment options, and the potential stigma associated with referral to services, such as mental health services.

C9: “I feel like trying to bridge that gap between what I know clinically works for trauma and I’m labelling it as trauma because pain is so intimately connected to that. But I guess coming back to what I was saying earlier and what you were explaining about is that there’s parts that I will never be capable of working with maybe on that. And I don’t know where else to refer those people to, I don’t know. I want to be able to give a recommendation but all I know is mental health and I know that doesn’t cut it and it’s so frustrating because that is what will help their pain. I don’t know if that’s, I don’t know how to answer that”.(Clinician at a metropolitan service)

In relation to communication about persistent pain, clinicians reported a desire to know the best ways to explain how pain occurs and what metaphors would be most useful for Aboriginal and Torres Strait Islander patients.

C14: “We use a lot of metaphors for educating people about pain and it could be good to kind of develop some ideas to make them a little bit more culturally appropriate and more engaging for the Indigenous population”.(Clinician at a metropolitan service)

Another area of need was the ability to comprehend communication practices that have a particular meaning in Aboriginal and Torres Strait Islander cultures, but that may not have the same meaning in the clinician’s culture (e.g., pauses during the conversation may for the clinician be a sign of disengagement or not understanding, but for the patient, be a natural way to interact). Difficulty comprehending such culture-specific cues was most often reported by clinicians who were less exposed to Aboriginal and Torres Strait Islanders cultures either because they had not grown up or were not educated in Australia. Clinicians indicated that they were aware that patients may use subtle verbal or non-verbal cues during a consultation that they, the clinicians, may not understand. Many patients have complex social needs and competing priorities, and thus not being able to identify the cues given by the patient can compromise the adherence to treatment and outcomes for a patient. This clinician was educated overseas and was unfamiliar with the expression “blackfella” and the discriminatory meaning.

C7: “For example this patient said, “I was bullied in school because I was a blackfella” so I was like, “Aaah so…”. I didn’t know, I didn’t know what that meant until later on when someone else said, “Why didn’t you pick up on that?” I was like, “I didn’t know””.(Clinician at a metropolitan service)


*Training Preferences*


Clinicians indicated they would like both cultural and communication training. The training should promote reflection on clinical practices, and during these reflections, the training should demonstrate how to blend biomedical knowledge with cultural approaches to improve clinical practice.

C9: “One size fits all is probably not going to address everyone’s needs. Because although we all work together we all have different training backgrounds and different experience backgrounds so I would think that a mixture of opportunities where people identify what particular skill they want to… skills or knowledge”.(Clinician at a metropolitan service)

C9: “So reflecting on perhaps some people are very confident in their communication style and clinical information gathering but are not sure you know “How do I bring in services practically?” all those sorts of things. Whereas other people might be like, “Oh well I’m really confident to ask Indigenous Liaison to come along but then how do I adapt my interpersonal skills?””.(Clinician at a metropolitan service)

Clinicians suggested training should be interactive and motivating, and this could be achieved through the combination of activities such as role-play, education, video examples and personalised feedback, and opportunities to apply the learnings.

C10: “You know in an online module and things like that, your click-through, I don’t think that I have got as much out of the online modules as compared to when I’ve sat in an interactive classroom set up by the gentleman sitting two seats over from me”.(Clinician at a regional service)

C9: “like it’s always nice to mix things together but obviously with learning it needs to be relevant to the role”. (Clinician at a regional service)

Clinicians reported that very often training is mainly theoretical and therefore difficult to translate to practice. Clinicians emphasized the importance of creating a relaxed environment, in which people are not coerced into activities or “being put in the spot”. Some clinicians indicated that training can make participants feel anxious that they would do something wrong.

C5: “I think it should be a kind of mixture because you’re going to have mixed personalities, introverted and extraverted and for some people complete interactive workshop is going to be boring for example for myself if it would be completely interactive and no didactic talk”. (Clinician at a regional service)

In general, clinicians agreed that shorter sessions were preferable. Some clinicians suggested, for example, a two-hour education session that would be supplemented with online video “refresher” courses in the months following (i.e., at 3 and 6 months post-training) to reinforce the practical application of the learning. Localised learning materials should be included for authenticity and resource material should be accessible electronically. Clinicians suggested that the follow-up training would benefit from having supervision or mentorship, perhaps even having their consultations supervised, with the goal of receiving feedback on their performance. The less preferable methods of training identified by clinicians were online learning, training that is solely didactic, and mandating interaction within training through compulsory activities.

## 4. Discussion

This study highlighted clinicians’ recognition of the importance of effective communication, as well as their perceived gaps in knowledge, ability, and confidence to communicate effectively with Aboriginal and Torres Strait Islander patients. Moreover, when invited to explore these gaps in the focus groups, clinicians clearly identified a lack of cultural knowledge and protocols as important barriers to communication. These two findings underlie the importance of support strategies to improve communication skills. Interestingly, clinicians in regional sites rated the importance of requisite knowledge to communicate effectively, and the ability and confidence to do so, more highly than their metropolitan-based counterparts.

In pain management, communication was found to be as therapeutic and effective as medication [37]. The importance of communication in pain assessment and care planning for Aboriginal and Torres Strait Islander patients was previously documented [7,16,20,21]. Evidence shows that clinician communication skills are essential to inform the patient and family about their disease, facilitate decision-making about treatment, and align treatment with the goals and values of patients and families [22]. Further, clinicians who described improvement in their active listening skills tended to gather more accurate information and promoted patient empowerment [20]. A study investigating factors limiting the effectiveness of communication between Aboriginal patients with end-stage renal disease and health care workers in the Northern Territory found that miscommunication is frequently unrecognised [38,39]. Identified sources of miscommunication included lack of patient control over the language, content and circumstances of interactions, absence of opportunity and resources to build a shared understanding, cultural and linguistic distance and lack of staff training in intercultural communication [20,21,38,40,41]. Internationally, a literature review on the experience, epidemiology and management of pain involving Indigenous populations (i.e., American Indian, Alaska Native and Aboriginal Canadian people) demonstrated that while there was modest literature about pain among these groups, poor communication between patient and provider emerged as an important need for pain management. There were differences between patients’ and clinicians’ communication about the pain, which may have led to miscommunication and misdiagnosis of what was being reported by the patient. For example, some “populations” could use distinctive communication practices in medical encounters, such as metaphors, that may be unfamiliar to and poorly understood by clinicians [6].

This study identified the needs of both cultural and communication aspects. Less than a third of clinicians rated their knowledge, ability and confidence to communicate effectively with Aboriginal and Torres Strait Islander patients as high. These findings are similar to those reported in a recent review of physical rehabilitation among Aboriginal and Torres Strait Islander patients, in which cultural development and the quality of communication were recommended as a way to realise equitable outcomes for Aboriginal and Torres Strait Islander people [13]. The cultural capability has been a major strategy to address health inequity and improve the quality of health care and outcomes among people from minority groups [40]. Cultural capability interventions were found to improve patient–clinician relationships, which in turn increases health service access [41].

Previous research shows that clinician competence in communication, in general, was found to be extremely variable, and a vast majority of clinicians learn to communicate through their practice, rather than through a formal curriculum, mentoring or evidence-based learning [22]. Communication was assumed as something that a clinician inherits or that can be acquired through a combination of passive exposure and trial and error [42,43]. Effective communication is actually essential in clinical settings and has a significant body of science behind it, it is not just a social skill to add on [44].

Communication between clinicians and Aboriginal and Torres Strait Islander patients has additional complexity; the level of clinicians’ knowledge of Aboriginal and Torres Strait Islander history and the social determinants of health, can either limit or alternatively, empower the patient–clinician relationship [45]. The higher importance, knowledge, ability and confidence ratings of clinicians at the regional site compared to the metropolitan sites may be due to the fact that there is a higher proportion of Aboriginal and Torres Strait Islander people per capita for the regional area (9%) than for the metropolitan areas (3%) [46], exposing clinicians in regional areas to a greater number of Aboriginal and Torres Strait Islander patients. In the focus groups, clinicians from the regional service also mentioned having a close relationship with the local cultural capability officers to guide them on effective communication and thus increasing their knowledge of how to effectively communicate with Aboriginal and Torres Strait Islander patients.

The historical, social and political context of Aboriginal and Torres Strait Islander communities, caused by colonisation and disempowerment, has impacted the availability of resources and access to services and this is an important factor in a shared understanding of the history and consequently meaning-making [19,45]. The lack of awareness and understanding of these historical, social and political factors within mainstream healthcare must be addressed to overcome communication barriers between Aboriginal and Torres Strait Islander patients and clinicians [45]. Further, clinicians in the focus groups acknowledged the value of being aware and paying attention to non-verbal cues during a consultation interaction. This is similar to what Little et al. (2015) [47] found with regard to verbal and non-verbal communication in a primary care setting. That is, patients could perceive the professional communication as poor, either distant or overly optimistic, depending on non-verbal elements of communication such as gestures, physical touch and social conversation [47].

In terms of communication styles, Aboriginal and Torres Strait Islander people indicated a preference for “yarning” styles of communication, a two-way dialogue incorporating careful listening, shared care planning and sharing of information of a non-clinical nature [25]. Clinicians require ongoing cultural education, localised to the region where they are working and relevant to their professional development and evolution of clinical practice [15]. For example, Aboriginal and Torres Strait Islander people have matters organised as women’s business and man business, the provision of services for female patients should preferably be provided by female clinicians to avoid feelings of shame [48,49]. Mixed education group sessions involving female patients with persistent pain in gynaecological parts could be a scenario that is not culturally sensitive and an unsafe provision of care [48].

Future interventions to improve communication between clinicians and Aboriginal and Torres Strait Islander patients should be patient-centred, observing local cultural principles of communication and evidence-based medicine [15,20]. Among the training preferences, consistent with previous studies, clinicians indicated that an interactive training format with the possibility of post-training feedback and follow-up or mentoring would be the most preferable training format [50,51]. A cross-sectional study exploring the learning preferences of 2500 general practitioners identified that 70% of participants indicated that interactive group discussions were their preferred learning format and that relevance to clinical practice was the main motivation for participation [52].

This study has several strengths. The study had a high response rate, with a majority of clinicians (71%) completing the cultural and communication awareness survey. Focus groups were facilitated by an Aboriginal researcher and a Torres Strait Islander cultural facilitator, who used their understanding of Aboriginal and Torres Strait Islander culture to explore clinicians’ perspectives. The focus groups were also conducted at each study site, ensuring that regional and metropolitan perspectives were captured. The qualitative data complemented information gathered about clinicians’ communication knowledge by identifying areas of need and preferences for training. Nevertheless, there were some limitations to the current study. This study has a cross-sectional design and provides a “snapshot” of the self-reported level of knowledge, ability and confidence of clinicians to communicate with Aboriginal and Torres Strait Islander patients and therefore is essentially descriptive. Not all occupations were represented in the focus groups and the findings may not be transferrable to clinical settings other than persistent pain services. Additionally, the involvement of an Aboriginal researcher and a Torres Strait Islander cultural facilitator in the focus groups may have constrained how clinicians discussed the importance of communication with Aboriginal and Torres Strait Islander patients. However, this is unlikely as the survey responses were congruent with the analysis of the focus group data and the survey was anonymous.

## 5. Conclusions

Clinicians identified training as an important strategy to address the unmet need for cultural and communication education. Clinicians described having limited knowledge and confidence in their ability to communicate with Aboriginal and Torres Strait Islander patients. Communication skills can be learned and training, in the form of a tailored intervention to support quality engagement with Aboriginal and Torres Strait Islander patients (and health outcomes), should combine cultural and communication aspects with biomedical knowledge and clinical skills. In pain management services, clinicians with an appreciation of cultural knowledge, a better understanding of the local context and the history of Aboriginal and Torres Strait Islander people, appeared to be essential to improve communication and quality of care for Aboriginal and Torres Strait Islander patients.

## Figures and Tables

**Figure 1 ijerph-19-01572-f001:**
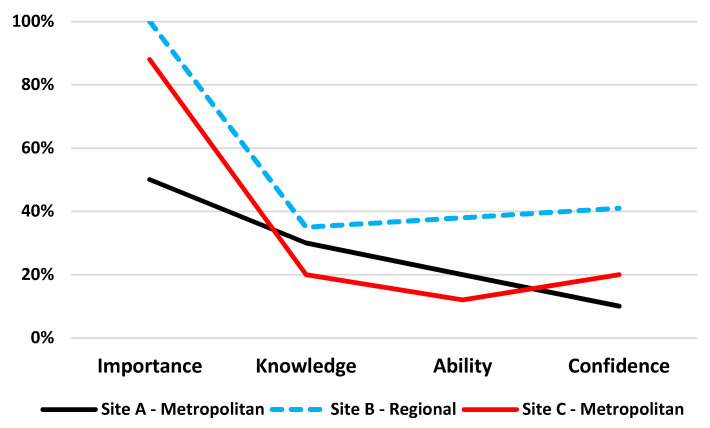
Proportion of clinicians who rated ‘*high*’ the importance, knowledge, ability and confidence to communicate with Indigenous Australian patients.

**Figure 2 ijerph-19-01572-f002:**
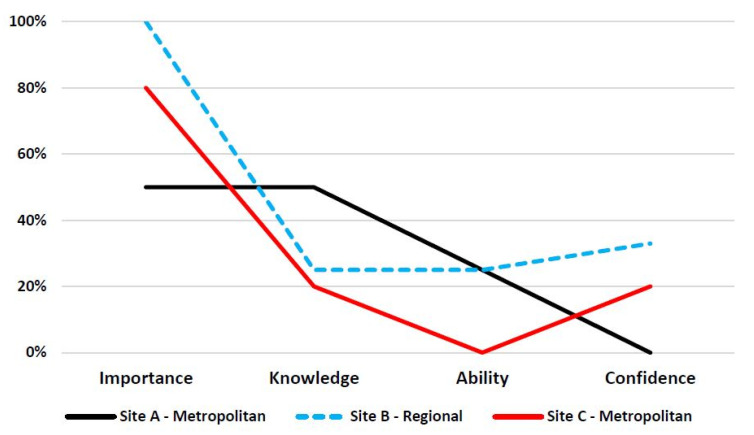
Proportion of medical doctors who rated *‘high’* the importance, knowledge, ability and confidence to communicate with Indigenous Australian patients.

**Figure 3 ijerph-19-01572-f003:**
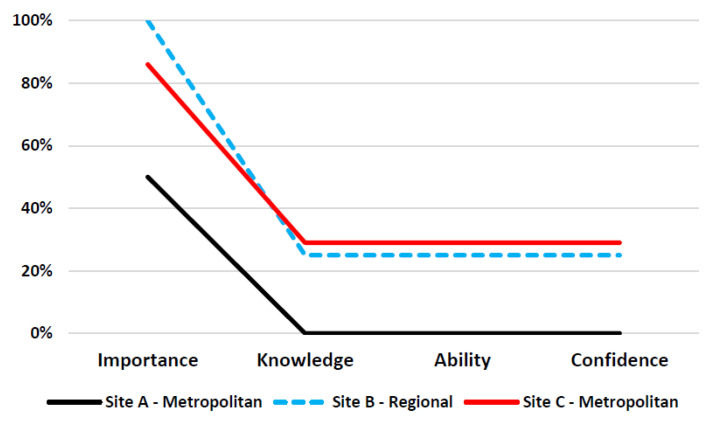
Proportion of nursing staff who rated ‘*high*’ the importance, knowledge, ability and confidence to communicate with Indigenous Australian patients.

**Table 1 ijerph-19-01572-t001:** Demographic characteristics and previous training information of clinicians who completed the cultural awareness and communication needs survey (CACNS) across the three study sites.

Demographic Characteristics and Previous Training	*N* = 64	
	*n*	%
Study site		
A—Metropolitan	10	16
B—Regional	29	45
C—Metropolitan	25	39
Age ^1^		
21–40 years	32	50
41–60 years	25	39
61–80 years	5	8
Sex		
Male	20	31
Female	43	67
Other	1	2
Indigenous status		
Indigenous	2	3
Non-Indigenous	62	97
Profession		
Medical doctors (pain medicine specialist, registrar, psychiatrist, GP senior medical officer)	26	41
Nursing Staff (registered nurse, clinical nurse, nurse navigator, enrolled nurse)	13	20
Physiotherapist	9	14
Psychologist	9	14
Pharmacist	1	2
Occupational Therapist	6	9
Previous cultural training		
Yes	47	73
No	17	27
If previous cultural training—type of training		
Queensland Health mandatory cultural awareness training/education packages	32	68
Cultural capability and safety lectures during university studies	7	15
Cultural awareness workshops (at Community Health Controlled Services and Remote Area Health Cops (RAHC))	5	11
Not specified/unknown	3	6

^1^ Two missing values.

**Table 2 ijerph-19-01572-t002:** Clinicians’ (*N* = 64) rating of the importance of training, knowledge, ability and confidence to communicate with Indigenous Australian patients.

Items	Study Site *	Score													
		Very Low		Low		High		Very High		Combined Low		Moderate		Combined High	
		1		2		4		5		1–2		3		4–5	
		*n*	(%)	*n*	(%)	*n*	(%)	*N*	(%)	*N*	(%)	*n*	(%)	*n*	(%)
Perceived *importance of communication training* when working with Aboriginal and Torres Strait Islander patients	A	-	(-)	2	(20)	4	(40)	1	(10)	2	(20)	3	(30)	5	(50)
	B	-	(-)	-	(-)	8	(28)	21	(72)	-	(-)	-	(-)	29	(100)
	C	-	(-)	1	(4)	9	(36)	13	(52)	1	(4)	2	(8)	22	(88)
	Total	-	(-)	3	(5)	21	(33)	35	(55)	3	(5)	5	(8)	56	(88)
Perceived *knowledge* of how to effectively communicate with Aboriginal and Torres Strait Islander patients	A	-	(-)	2	(20)	3	(30)	-	(-)	2	(20)	5	(50)	3	(30)
	B	-	(-)	1	(3)	8	(28)	2	(7)	1	(3)	18	(62)	10	(35)
	C	1	(4)	7	(28)	5	(20)	-	(-)	8	(32)	12	(48)	5	(20)
	Total	1	(2)	10	(16)	16	(25)	2	(3)	11	(17)	35	(55)	18	(28)
Perceived *ability* to communicate with Aboriginal and Torres Strait Islander patients	A	-	(-)	-	(-)	2	(20)	-	(-)	-	(-)	8	(80)	2	(20)
	B	-	(-)	-	(-)	10	(35)	1	(3)	-	(-)	18	(62)	11	(38)
	C	-	(-)	4	(16)	3	(12)	-	(-)	4	(16)	18	(72)	3	(12)
	Total	-	(-)	4	(6)	15	(23)	1	(2)	4	(6)	44	(69)	16	(25)
Perceived *confidence* to communicate with Aboriginal and Torres Strait Islander patients	A	-	(-)	1	(10)	1	(10)	-	(-)	1	(10)	8	(80)	1	(10)
	B	-	(-)	-	(-)	11	(38)	1	(3)	-	(-)	17	(59)	12	(41)
	C	-	(-)	4	(16)	5	(20)	-	(-)	4	(16)	16	(64)	5	(20)
	Total	-	(-)	5	(8)	17	(27)	1	(2)	5	(8)	41	(64)	18	(28)

* Study sites A and C are metropolitan and study site B is regional.

**Table 3 ijerph-19-01572-t003:** Number of clinicians who participated in the focus group by occupation.

Occupation	Number	%
Nurse Navigator	1	5
Occupational Therapist	4	20
Pain Specialist	6	30
Physiotherapist	2	10
Psychiatrist	1	5
Psychologist	3	15
Registrar	2	10
GP Senior Medical Officer	1	5
Total	20	100

## Data Availability

The data presented in this study are available on request from the corresponding author. The data are not publicly available due to restrictions regarding the Ethical Committee Institution.

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
