# Peer review of "Clinician Perspectives of Communication with Aboriginal and Torres Strait Islanders Managing Pain: Needs and Preferences"

_ijerph, 2022, doi:10.3390/ijerph19031572_

Round 1
Reviewer 1 Report
This article reports on a very important topic: communication between clinicians and Aboriginal and Torres Strait Islander patients. Given the wide gap in health status between Indigenous and non-Indigenous Australians and the lack of equal access to primary health care in regional and remote communities, clear and culturally appropriate health communication is clearly of vital importance in improving the unacceptable health situation that exists for Aboriginal and Torres Strait Islander peoples. The article investigates this domain by focusing on the self-reported communication needs and training preferences of clinicians.
The introduction is clear and well-structured, providing background and a review of the literature relating to persistent pain management, health services in Queensland, and some of the communication challenges that healthcare providers face.
In the materials and methods section, the study design and data collection was fairly clear, (using a short survey and facilitated focus groups). It would have been useful to have more comprehensive demographic information recorded, particularly about the amount of time participants had spent living and working in the communities that they were based. With respect to the focus groups, further information could be provided about the size and make-up of the focus groups (e.g. in each group were there a mix of health professionals or they were grouped according to profession? Did either of the 2 Indigenous clinicians who took part in the survey also take part in the focus groups? Did the participants in each of the focus groups already know each other/ work together?).
In the results and discussion sections there is substantial discussion of non-verbal cues with respect to successfully navigating communication practices, but little discussion of verbal cues, language use, language background of patients, and how clinicians approach and navigate this. It would be interesting and helpful to know if the participants discussed concerns relating to this, or to acknowledge this area if it didn’t come up in the topics raised during the focus groups, as it’s clearly an area that can cause communication difficulties between different service providers/groups of people/etc.
See the attached document for more specific comments.

Author Response
Thank you for your constructive and helpful feedback.
Kind regards

Reviewer 2 Report
Dear Authors,
Congratulations on a very interesting and extremely useful piece of research.
Contentwise I think your manuscript is impeccable. I just spotted the occurrence of double spaces along the whole text.
Best regards,
Author Response
Thank you, the whole team enjoyed your comments.
Kind regards

Reviewer 3 Report
An interesting article with strengths and limitations that you have pointed out yourselves. Before this goes into print, check for minor imperfections in the text. Here are some:
47: Persistent pain (i.e., pain lasting > 3months) commonly involves a combination of > 3 months (space)
88: ‘cultural capability refer to skills, knowledge and behaviours that are required to plan, … > refers
135 Analysis of data from Aboriginal and Torres 134 Strait Islander consumers’ is the focus … > why genitive? missing noun after consumers’?
308: it’s okay to ask if somebody identifies[as an Aboriginal and Torres Strait Islander ] > missing space resp. superfluous space
322: made by one of the facilitator about culture and illness > facilitators
346 and 353: C9: quotation marks are missing - also missing in lines 387 & 391; and in line 400
385: knowledge..’
467: Communication has been assumed as something that a clinician inherit or that can be acquired > inherits
497: the provision of services for female patients should be preferable be provided by female clinicians > should preferably be
513: The focus group were > The focus groups were
525: responses, in which were anonymous, > unclear
Author Response
Thank you for your helpful feedback.
Kind regards
